# Improved Accuracy in Predicting the Best Sensor Fusion Architecture for Multiple Domains

**DOI:** 10.3390/s21217007

**Published:** 2021-10-22

**Authors:** Erik Molino-Minero-Re, Antonio A. Aguileta, Ramon F. Brena, Enrique Garcia-Ceja

**Affiliations:** 1Instituto de Investigaciones en Matemáticas Aplicadas y en Sistemas—Unidad Yucatán, Universidad Nacional Autónoma de México, Sierra Papacal, Yucatán 97302, Mexico; erik.molino@iimas.unam.mx; 2Facultad de Matemáticas, Universidad Autónoma de Yucatán, Anillo Periférico Norte, Tablaje Cat. 13615, Colonia Chuburná Hidalgo Inn, Mérida, Yucatán 97110, Mexico; aaguilet@correo.uady.mx; 3Tecnologico de Monterrey, Av. Eugenio Garza Sada 2501 Sur, Monterrey, NL 64849, Mexico; 4Optimeering AS Tordenskioldsgate 6, 0160 Oslo, Norway; e.g.mx@ieee.org

**Keywords:** sensor fusion, classification, SFFS, metadata, statistical signature

## Abstract

Multi-sensor fusion intends to boost the general reliability of a decision-making procedure or allow one sensor to compensate for others’ shortcomings. This field has been so prominent that authors have proposed many different fusion approaches, or “architectures” as we call them when they are structurally different, so it is now challenging to prescribe which one is better for a specific collection of sensors and a particular application environment, other than by trial and error. We propose an approach capable of predicting the best fusion architecture (from predefined options) for a given dataset. This method involves the construction of a meta-dataset where statistical characteristics from the original dataset are extracted. One challenge is that each dataset has a different number of variables (columns). Previous work took the principal component analysis’s first *k* components to make the meta-dataset columns coherent and trained machine learning classifiers to predict the best fusion architecture. In this paper, we take a new route to build the meta-dataset. We use the Sequential Forward Floating Selection algorithm and a *T* transform to reduce the features and match them to a given number, respectively. Our findings indicate that our proposed method could improve the accuracy in predicting the best sensor fusion architecture for multiple domains.

## 1. Introduction

The combined use of multiple sensors, either similar or different, to take measurements of a given phenomenon is a good way to compensate for the weaknesses (for instance, lack of precision, malfunction of a sensor, uncertainty, and limited spatial coverage [1]) of some sensors with the help of others [2,3,4]. The combination of several sensors has the advantage of increasing reliability, robustness, resolution, precision, and other desirable properties while decreasing uncertainty and ambiguity [5].

The multi-sensor strategy has gained such prominence that many fusion methods have been described in the literature such as Feature Aggregation (FA) [3,6], Voting (Vot) [7], Multi-view stacking (MVS) [8], and AdaBoost (AB) [9]. The problem is that there are now so many methods that it is hard from the outset which of the many possible fusion configurations to choose for a specific set of sensors [10], mainly because none of the proposals we found explains why it works for the particular set of sensors they use (we suspect many authors just use the methods they are familiar with). We call the sensor fusion methods “architectures” when we want to emphasize the structural differences, such as several levels of fusion or the use of similar or different sensors.

One way to mitigate the problem of fusion architecture proliferation is to be able to foretell the optimal fusion strategy for a specific given dataset. To our knowledge, only two studies have addressed this problem: the proposal by Aguileta et al. [11] and the one of Brena et al. [12]. Aguileta et al. [11] introduced a machine learning approach laid on a meta-dataset of statistical signatures (used as a vector of features) to perform this task for human activity recognition (HAR), where they achieved a 90% precision in predicting the best fusion architecture among eight (which used the previously mentioned methods: FA, Vot, MVS, and AB) for a human activity dataset. Furthermore, Brena et al. [12] extended this approach to identify (with 91% accuracy) the optimal fusion architecture (among these eight fusion architectures) for a specific dataset from other fields besides HAR, including chemical gas classification [13] and grammatical facial expressions (GFE) classification [6]. A key aspect of this extended approach was the use of the principal component analysis (PCA) [14] method to construct a statistical signature (SS) meta-dataset. The statistical signature will be referred to as SS from now on, and it is essentially a vector of features, each of which is a statistical trait of a dataset in our collection. Details on how to calculate SS are presented in Section 3.2.

Overall, the method consists of building a “meta-dataset” where each row is built by trying several fusion architectures for different sensor combinations, then recording the name of the best performing architecture for a given combination, which will be later used as the class label for that row. Each row in the meta-dataset is completed by extracting statistical features (SS) from the “source” datasets derived from the different sensor combinations. Then, a classifier is trained with the meta-dataset to predict the name of the best architecture based on the input SS.

One of the challenges to be solved is that since each dataset is reduced to a single SS row in the meta-dataset, it is extremely difficult to construct meta-datasets of more than a few dozen rows. Further, there is the problem that if the meta-dataset contains rows from different domains (such as HAR, gas identification, and GFE recognition, as in this work), we have to align the columns in such a way that the columns are compatible, both in their number and also in their meaning, regardless of the considered domains, and eventually additional ones. The previous solution [12] used PCA to reduce the SS columns (used as features) to combine them, regardless of the domain, where one column corresponds to the first PCA component, the next to the second component, and so on.

Although the approaches described above have addressed the problem of identifying the appropriate sensor fusion for a specific dataset, there is still a precision and accuracy gap to fill (around 9% accuracy). Furthermore, other commonly used methods for reducing features (such as the Sequential Forward Floating Selection (SFFS) [15] algorithm) were left unexplored.

In this work, we describe a method that improves the accuracy achieved by Brena et al. [12] through the SFFS algorithm and a data transformation matrix *T* (see Equation (Equation 8)) to build a meta-dataset from the SS of the three mentioned domains. The idea is to reduce the dimensions of the SS datasets using the SFFS algorithm. Then, to match the dimensions of such reduced SS with a given size, using the proposed transformation *T*. Afterwards, to build a meta-dataset, row by row, combining these reduced SS set. Finally, to train a machine learning algorithm so that it learns to predict the best fusion architecture (using this new meta-dataset), within the eight methods mentioned above, for a particular dataset from any of the domains.

The remaining of this paper is organized as follows. First, a review of the state-of-the-art is presented, regarding sensor-data fusion techniques. Then, we describe the methodology for choosing the most effective fusion method using the SFFS algorithm and the proposed data transformation. Next, we describe the datasets, their assembly, and the experimental setup, followed by the results and a discussion. We end with our conclusions.

## 2. Background on Sensor Fusion Techniques

Here, we review the previous works in the area of multi-sensor fusion techniques (ranked by level of data abstraction) and feature selection (using the SFFS algorithm), respectively.

### 2.1. Multi-Sensor Fusion

Multi-sensor fusion was conceptualized in the 1970s by the US Naval as a technique to boost the Soviet Navy’s motion detection system’s accuracy, among other military problems [16]. Over time, this strategy moved to the civil context and is currently being used in diagnostics in medicine, robotics, video processing, and smart buildings [17].

Furthermore, this strategy has been so successful that many fusion methods have been proposed to such an extent that various proposals have emerged to organize them. One of them classifies fusion architectures into three classes according to the degree of abstraction of data processing (fusion of the data, the features, or the decisions) [18,19]. However, not all fusion methods fit into one of these categories because they combine two of them. These methods have been classified in the category of two-level fusion [10]. We briefly explain these categories below and provide some examples.

*Fusion of the data*: sensors’ unprocessed data are grouped into sets with the idea of obtaining a higher quality (accurate, informative, and synthetic) in the grouped data than when it is not grouped [20]. Raw Data Aggregation (RDA) [21] and Time-lagged Similarity Features [22] are methods that fit into this category [10].

*Fusion of the features*: Combines the features of the sensor data to create a large feature vector from which a classifier can learn to identify patterns [23,24]. FA, Temporal Fusion (TF) [25], and the Data Fusion Location algorithm [18] are examples of feature-level fusion.

*Fusion of the decisions*: A final decision (a target class or tag) is made after combining the results (considered intermediate) from several classifiers or decision processes [26]. This type of fusion can be seen in the dynamic multi-sensor data fusion approach based on evidence theory and weighted ordered weighted averaging operator [27], Vot, and MVS.

*Fusion at two levels*: Two of the fusion styles described above are combined by fusion architectures proposed in the literature [10]. For example, some methods merge data and features, such as combining TF (feature-level) with RDA (data-level) [21]. Other methods also merge features and decisions, such as the combination of Genetic Algorithm-Based Classifiers Fusion (decision-level) with Feature Combination [28] (feature-level) (GABCF-FC) [29].

These fusion methods were developed and tested on data from specific sensors without further evaluation. Consequently, given a set of sensors, we do not know which fusion method to choose. This problem was addressed by Brena et al. [12], where a method to predict the best fusion method for a given set of sensors was proposed. The contribution of this paper is then a proposal to improve the classification accuracy over Brena et al. work.

### 2.2. The SFFS Algorithm

Feature selection aims to take a larger collection of features (which we call *D*) and choose a subgroup of features (which we call *d*) from this large group, where d<D. The performance of a machine learning algorithm using subset *d* should not significantly decrease relative to the achieved performance of that algorithm when using set *D*. The selection of features can indeed be viewed as a search issue to find the best feature subgroup according to a provided measure. A suitable criterion function calculates this measure. From this problem emerges the necessity of developing computationally feasible procedures that avoid the exhaustive search, although their result is a feature set that may be suboptimal [15]. Although there have been several feature selection algorithms proposed in the literature (such as the Sequential Backward Selection (SBS) method [30], Sequential Forward Selection (SFS) method [31], Sequential Backward Floating Selection (SBFS) algorithm [15], to name a few), in this work, we use the SFFS algorithm as both its performance and computational efficiency are outstanding compared to other methods. Here is a formal description of the SFFS algorithm, taken from Pudil et al. [15].

Let $X_{k}=\left\{x_{1},x_{2},..,x_{k}\right\}$ denote the group of *k* features from the group $Y=\{y_{1},y_{2},...,y_{D}\}$, where *D* is the total number of features and $x_{i}\in Y$ for i=1,2,...,k. The individual significance $S_{o}\left(y_{i}\right)$ is defined as the value $J(y_{i})$ of the criterion function for deciding which features to use, where $y_{i}$ is the *i*th feature, for i=1,2,...,D. The significance $S_{k-1}\left(x_{j}\right)$ of the feature $x_{j}$, with j=1,2,...,D, in the group $X_{k}$ is specified in Equation (Equation 1).
(1)$$S_{k-1}\left(x_{j}\right)=J\left(X_{k}\right)-J\left(X_{k}-x_{j}\right)$$

The relevance $S_{k+1}\left(f_{j}\right)$ of the feature $f_{j}\in Y-X_{k}=\left\{f_{1},f_{2},...,f_{D-K}\right\}$, where $f_{i}\in Y$ and $f_{i}\neq x_{l}$ for all $x_{l}\in X_{k}$, is defined in Equation (Equation 2).
(2)$$S_{k+1}\left(F_{j}\right)=J\left(X_{k}+f_{i}\right)-J\left(X_{k}\right)$$

The feature $x_{j}\in X_{k}$ is the most relevant (best) feature in such set $X_{k}$ if
(3)$$\begin{array}{c} S_{k-1}\left(x_{j}\right)=\max_{1\leq i\leq k}S_{k-1}\left(x_{i}\right)\\ \Rightarrow J\left(X_{k}-x_{j}\right)=\min_{1\leq i\leq k}J\left(X_{k}-x_{i}\right) \end{array}$$

The feature $x_{j}\in X_{k}$ is the least relevant (worst) feature in such set $X_{k}$ if
(4)$$\begin{array}{c} S_{k-1}\left(x_{j}\right)=\min_{1\leq i\leq k}S_{k-1}\left(x_{i}\right)\\ \Rightarrow J\left(X_{k}-x_{j}\right)=\max_{1\leq i\leq k}J\left(X_{k}-x_{i}\right) \end{array}$$

The feature $f_{j}\in Y-X_{k}$ is the most relevant (best) feature regarding $X_{k}$ if
(5)$$\begin{array}{c} S_{k+1}\left(f_{j}\right)=\max_{1\leq i\leq D-k}S_{k+1}\left(f_{i}\right)\\ \Rightarrow J\left(X_{k}+f_{j}\right)=\max_{1\leq i\leq D-k}J\left(X_{k}+f_{i}\right) \end{array}$$

The feature $f_{j}\in Y-X_{k}$ is the least relevant (worst) feature regarding to $X_{k}$ if
(6)$$\begin{array}{c} S_{k+1}\left(f_{j}\right)=\min_{1\leq i\leq D-k}S_{k+1}\left(f_{i}\right)\\ \Rightarrow J\left(X_{k}+f_{j}\right)=\max_{1\leq i\leq D-k}J\left(X_{k}+f_{i}\right) \end{array}$$

Assume that the criterion function $J(X_{k})$ was used to create a set $X_{k}\subset Y$ and that $J(X_{i})$ values are known and stored for all subsets of size i=1,2,...,K−1. Then, the SFFS algorithm follows the stages outlined below.

Stage 1: choose the feature $x_{k+1}\in Y-X_{k}$, according to the SFS method, to construct the feature group $X_{k+1}$. Hence, $X_{k+1}=X_{k}+x_{k+1}$, with $x_{k+1}$ being the most relevant characteristic according to the group $X_{k}$—it is the Inclusion stage.Stage 2: from the group $X_{k+1}$, pick the least relevant feature $x_{k+1}$. Therefore, $J\left(X_{k+1}-x_{k+1}\right)\geq J\left(X_{k+1}-x_{j}\right)$, ∀j=1,2,...,k. So, group k=k+1 and go back to Stage 1. If $x_{r}\in X_{k+1}$, for 1≤r≤k, is the least relevant feature, then $J\left(X_{k+1}-x_{r}\right)>J\left(X_{k}\right)$. Therefore, the new group $X_{k}^{\prime }=X_{k+1}-x_{r}$ must be formed. Notice that now $J\left(X_{k}^{\prime }\right)>J\left(X_{k}\right)$. If k=2, then put $X_{k}=X_{k}^{\prime }$ and $J\left(X_{k}\right)=J\left(X_{k}^{\prime }\right)$ and go back to Stage 1, otherwise go to Stage 3. This step is called Conditional Exclusion.Stage 3: choose the least relevant feature $X_{s}\in X_{k}^{\prime }$. If $J\left(X_{k}^{\prime }-x_{s}\right)\leq J\left(X_{k-1}\right)$ then put $X_{k}=X_{k}^{\prime }$, $J\left(X_{k}\right)=J\left(X_{k}^{\prime }\right)$, and go back to Stage 1. If $J\left(X_{k}^{\prime }-x_{s}\right)>J\left(X_{k_{1}}\right)$ then the new group $X_{k-1}^{\prime }=X_{k}^{\prime }-x_{s}$ must be formed. Put k=k−1. If k=2, then establish $X_{k}=X_{k}^{\prime }$ and $J\left(X_{k}\right)=J\left(X_{k}^{\prime }\right)$ and go back to Stage 1, otherwise repeat Stage 3. This is what follows the Conditional Exclusion stage.

The parameters are initialized (k=0 and $X_{o}=\emptyset$). The SFS approach is employed as long as the feature set does not exceed cardinality two. Then the algorithm proceeds with Stage 2.

## 3. Method

Figure 1 presents the approach proposed by Brena et al. [12] that predicts the most suitable fusion architecture for a given set of data (e.g., HAR [32], gas detection [13], or GFE recognition [6]). In this section, we briefly explain this approach, first, in Section 3.1 the process for choosing the most effective fusion architecture is described. Then, in Section 3.2, the details of the statistical signature dataset stage are explained, as this is the core of our proposal, and we are following a different approach than the one described by the original authors. Furthermore, finally, in Section 3.3, the prediction of the optimal fusion architecture step is briefly described. For more details, the reader can consult the work of Brena et al. [12].

### 3.1. Choosing the Most Effective Fusion Architecture

In the upper part of Figure 1, we can see how the tag (that is, which one is the most appropriate fusion architecture for a specific dataset) is obtained. Through a statistical analysis, we find the best sensor data fusion architecture out of eight predefined ones for a given dataset [11].

In the following, we give a high-level explanation of the eight fusion architectures we are considering in this study (bear in mind that this small collection is by no means comprehensive, though the methods we are presenting are in principle applicable to many other fusion architectures):

**Aggregation (Agg)**. This architecture is a simple feature-level one, so much so that we take it as the baseline method. The idea is to just take every available feature from every sensor. It comprises two steps: the first one is to combine the features derived from the considered dataset by column; the second step is to train a classifier—for instance Random Forest (RFC) [33]—with these features in order to identify the actual labels of the dataset.

**Voting with shuffled features (VotWSF)**. This architecture operates at the level of decisions and comprises five phases: (1) we start by combining by column the features taken from the dataset under consideration; (2) we shuffle the features; (3) we divide them into three sections of equal size; (4) we run a standard classifier (like RFC) on each of the three partitions; and finally (5) we vote (using Vot) on the decision of each partition to reach a final decision.

**Voting without shuffled features (VotWoSF)**. This architecture is similar to the one before it (VotWSF). The only difference is that phase two (to shuffle the features) is skipped. The ones left steps are much like the ones before them.

**Voting with three classifiers for all features (VotWTCAF)**. This architecture uses fusion at the decision and feature levels, and works as follows: (1) obtain the features of the dataset being considered; (2) use three different classifiers—we have used Logistic Regression (LR) [34], Decision Tree (CART) [35] and RFC—with all the features; (3) train the classifiers and assemble the final result using Vot.

**Multi-View Stacking with shuffled features (MVSWSF)**. This process fuses the data at a decision-level. It comprises the following phases: (1) acquiring the features derived from the dataset being considered, and, taking the features as columns, shuffle them, and then break them into three sections; (2) take three instances of a standard classification algorithm (e.g., Random Forest) as the first-level learner; (3) train the base learners instances with some of those three feature portions and aggregate the decisions of those instances by column; (4) take a base ML classifier (RFC for instance) as a meta-learner to learn from the base learners’ decisions so that it will be able to identify the labels of the given dataset.

**Multi-View Stacking without shuffled features (MVSWoSF)**. This architecture differs from the previous one (MVSWSF) in that the features are not shuffled. The remaining steps, on the other hand, remain unchanged.

**Multi-View Stacking with three classifiers for all features (MVSWTCAF)**. This architecture, which uses two forms of fusion (feature and decision levels), requires four phases: (1) generate and aggregate the characteristics obtained from the dataset being considered by columns—which is a feature-level fusion; (2) determine three classification techniques (CART, RFC, or LR) as base learners; (3) train these aggregated features for each base learners and combine the decisions by column; and (4) take a standard ML predictor (RFC, for instance), which will play the role of a meta-learner, in a decision-level fusion, and it will be trained with the aggregated decision of step (3), so that it predicts the classes of the given dataset.

**AdaBoost with RFC (ABWRFC)**. The four phases in this architecture, which uses various forms of fusion (e.g., decision/function level), consist of: (1) integrating the features generated from the considered dataset column-wise (merging feature at level); (2) determining a predictor (for example, RFC); (3) designing the Adaboost classifier as a high-level method (to perform decision-level merging); and (4) training AB with the features extracted in phase 1 so that it identifies the labels recorded in the given dataset.

We briefly describe the statistical analysis below. It relies on the Friedman’s rank test [36] and Holm’s test [37] to find substantial discrepancies concerning accuracy (if any) among the Aggregation (designated as the reference when the goal is the comparison) and each of the fusion architectures [11]. Some fusion architectures (different from Agg) exhibit a substantial advantage over the accuracy of Agg. The best of the examined fusion architectures is taken as such, as we explained above.

### 3.2. Statistical Signature Dataset

Here, we construct a meta-dataset with labeled statistical signatures derived from each dataset from one of the domains under consideration, following a different path than the original author (Brena et al. [12] ). To build this meta-dataset, we use the SFFS algorithm and the T transform (see Equation (Equation 8)), rather than PCA as the original author. Next, we show the method for generating this collection of meta-data (see Figure 2, which presents the general structure of the mentioned method; refer to Figure 3, Figure 4, Figure 5 and Figure 6 for method details), which is a simplified set of meta-data as it incorporates the SS of datasets from various domains.

First, a SS of the characteristics of each dataset from each domain is generated. This is shown in Figure 2. The method for generating the SSs is to obtain the mean, standard deviation, the minimum value, maximum value, and percentiles 75th, 50th, and 25th, for each of the datasets’ features [11].

Let a given dataset be characterized as a matrix *A*, with *S* rows (samples) and *F* columns (characteristics). Let $a_{ij}$ represent an entry in *A*, with i=1,2,...,S and j=1,2,...,F. Thus, *A* can be introduced as a collection of vectors $AV=\{f_{1},f_{2},..,f_{F}\}$, with $f_{1}={\left[a_{11},a_{21},...,a_{S1}\right]}^{T}$, $f_{2}={\left[a_{12},a_{22},...,a_{S2}\right]}^{T}$, ..., and $f_{F}={\left[a_{1F},a_{2F},...,a_{SF}\right]}^{T}$.

Thus, AV SS corresponds to the set:(7)$$\begin{array}{c} SSAV=\{mean\left(f_{1}\right),std\left(f_{1}\right),max\left(f_{1}\right),min\left(f_{1}\right),P_{25th}\left(f_{1}\right),P_{50th}\left(f_{1}\right),P_{75th}\left(f_{1}\right),\\ mean\left(f_{2}\right),std\left(f_{2}\right),max\left(f_{2}\right),min\left(f_{2}\right),P_{25th}\left(f_{2}\right),P_{50th}\left(f_{2}\right),P_{75th}\left(f_{2}\right),...,mean\left(f_{F}\right),\\ std\left(f_{F}\right),max\left(f_{F}\right),min\left(f_{F}\right),P_{25th}\left(f_{F}\right),P_{50th}\left(f_{F}\right),P_{75th}\left(f_{F}\right)\} \end{array}$$

Then, combine the SS derived from each dataset by row in the SS combination step, by domain [11]. Next (unlike the original authors [12], who use PCA to reduce the SS dimension), by domain, decrease the SS dimension using the SFFS algorithm (in the generalization step) and match it to a given number using the *T* transformation (see Equation (Equation 8)).

We used this algorithm since it is commonly used to minimize a function vector’s dimensions [38]. In the first part (forward selection), SFFS adds the characteristic (feature) of the characteristic space (feature space) that contributes to the maximum performance improvement for the subset of characteristics, determined by the following criterion: the characteristic that, if applied to this subset, is correlated with the best performance of the classifier. SFFS excludes a characteristic in the second component (floating selection) if an increase in output is achieved by the resulting subset. If the number of elements is equal to or cannot be increased in this subset, return to the first part; otherwise, repeat it. Redo parts one and two till you have the desired number of items [15]. Then, from the SFFS algorithm, the reduced SS per domain (RSSD) is obtained: $RSSD_{j}=\left\{d_{j}e_{1},d_{j}e_{2},...,d_{j}e_{nte_{j}}\right\}$, where j={1,2,...,D}, *D* is the total number of domains, $d_{j}e_{l}$ is the SS element *l* of domain *j*, and $nte_{j}$ is the total number of elements in domain *j*. We define $T:R^{nte_{j}}\to R^{k}$ in Equation (Equation 8), where $k=\sum_{i=1}^{D}\left[nte_{i}\right]$ and $O_{mxn}$ is a zero matrix of *m* rows by *n* columns.
(8)$$T\left(RF\right)=\left\{\begin{array}{cccc} \left(RSSD_{1},O_{1x(k-nte_{1})}\right) & \mathrm{if}RF=RSSD_{1}, & \mathrm{where} & RSSD_{1}\in R^{nte_{1}}\\ \left(O_{1x(nte_{1})},RSSD_{2},O_{1x(k-nte_{1}-nte_{2})}\right) & \mathrm{if}RF=RSSD_{2}, & \mathrm{where} & RSSD_{2}\in R^{nte_{2}}\\ \left(O_{1x(nte_{1}+nte_{2})},RSSD_{3},O_{1x(k-nte_{1}-nte_{2}-nte_{3})}\right) & \mathrm{if}RF=RSSD_{3}, & \mathrm{where} & RSSD_{3}\in R^{nte_{3}}\\ \cdots \\ \left(O_{1x(k-nte_{D})},RSSD_{D}\right) & \mathrm{if}RF=RSSD_{D}, & \mathrm{where} & RSSD_{D}\in R^{nte_{D}} \end{array}\right.$$

Subsequently, at the point of reduced SS labeling, the reduced SS shall be labeled with the best associated fusion architecture achieved at the preceding step. To create the final dataset of SS, the labeled and reduced SS from each dataset are stacked row-wise.

### 3.3. Predicting the Optimal Fusion Architecture

For a given data collection, this step attempts to predict the optimal fusion architecture. In order to do so, train a classifier (e.g., a Random Forest) with the SS dataset generated in the preceding step using k-fold cross-validation to identify the optimal fusion architecture for the target dataset. We explain the specifics of how we perform training and prediction in the following section.

## 4. Datasets Assembly and Experimental Set-Up

This section presents the experimental context, describing the datasets used, the features extracted, and the procedure followed to test our proposal. This procedure consists of two steps, that is, to select the best setup of fusion architectures, and to predict the best configuration of fusion architectures. In the first step, we select the configuration of the best fusion architecture after conducting a comparative analysis of the different data integration architectures. The second step consists of training a classifier that estimates the best fusion method, of which there are eight possible options for a set of data not used in training by cross-validation of k-folds. For instance, the dataset can be one of the datasets we have considered: simple human activity (SHA) [32], gases, or grammatical facial expressions (GFEs).

### 4.1. Datasets Configuration

To test the proposed method, we make use of several SHA, gas, and GFE datasets, and organize them into a new, larger set of 116 datasets, as follows.

#### 4.1.1. Simple Human Activity Dataset

We generate 40 SHA datasets from six different repositories [39,40,41,42,43,44], which are well known in the literature [11]. All of them use different sensors located around the body, being the most common, accelerometers and gyroscopes. The procedure to assemble the 40 SHA datasets is simple and consists of pairing two sensors’ data (accelerometer and gyroscope) from each dataset [12].

(1)We used the Multimodal Human Action Dataset (MHAD) from the University of Texas [39] to construct a set of data pairs. The original data were recorded using a Microsoft Kinect sensor and movement sensors, as 3-axis accelerometers (Acc) and gyroscopes (Gyr). It includes activities from 8 subjects and 27 sportive actions, each repeated four times, like swipe, clap, through, boxing, etc.(2)From the Opportunity Activity Recognition set [40], we generated ten datasets using the paired combination of sensors proposed in Brena et al. [12] (see Table 1). The original data contain 2477 instances of daily activity acquired through multimodal sensors (mainly Acc and Gyr), placed on the body of four subjects, while they performed four different physical activities including standing, walking, sitting, and lying down.(3)From the Physical Activity Monitoring for Aging People (PAMAP2) database [41], we generated seven sets of data from sensor pairs in the same way as Brena et al. [12] did it for this set (see Table 2). The original dataset consists of activity from inertial sensors (mainly Acc and Gyr), from nine subjects performing 18 actions, as lie, sit, stand, walk, run, etc.(4)From the Mobile Health dataset (MHealth) set [42], we generated four sets of data pairs considering the sensor configuration that Brena et al. [12] used for this set (see Table 3). The original dataset consists of activity from 10 subjects performing 12 actions, as lie, walk, climb stairs, waist end, etc.(5)From the Daily and Sports Activities (DSA) set [43], we generated 17 sets of data pairs using the same sensor combination used in previous work [12] for this set (see Table 4). The original data consist of sports activities from eight subjects performing 19 actions, for 5 minutes each, as sit, lie, climb stairs, stand, walk, etc.(6)From the Human Activities and Postural Transitions (HAPT) set [44], we generated one set of data pairs. The original data consist on daily activity from 30 subjects performing 12 activities, wearing a smart cell phone on the waist, as walk, climb upstairs, climb downstairs, stand to sit, lie, etc.

#### 4.1.2. Gas Datasets

The Gas Sensor Array Drift (GSAD) database [13], is commonly utilized for gas identification. The information corresponds to data acquired by 16 sensors during 36 months and indicates the concentration of six different types of gases (such as acetone, ethane, ethylene, etc.). From the data of month 36, we generate 36 sets of data pairs, matching the array of 16 sensors, according to the procedure proposed in Brena et al. [12].

#### 4.1.3. Grammatical Facial Expressions Dataset

From the Grammatical Facial Expressions (GFE) dataset [6], we generate 40 sets of data. The original data consist of 18 videos gathered using a Microsoft Kinect device, which recorded facial expressions from nine emotions, tagged into two classes each as positive (P) and negative (N). On each frame, 100 face landmarks are located as *x*, *y*, and *z*, for width, height, and depth, respectively. These cover each eye, iris, and eyebrow, nose, nose tip, mouth, face contour, etc. From these points, we extract the same points that Brena et al. [12] did to construct 40 sets. Due to differences in the number of observations in each class, we balance the sets by subsampling the majority class by randomly eliminating observations through the process of resampling the majority class without replacement and matching the minority class [45,46].

### 4.2. Feature Extraction

For each of the 40 SHA data pairs 16 statistical characteristics are extracted. The process consists of dividing each pair into three-second segments, without overlapping, and calculating several values including the mean, standard deviation, the maximum, correlation, the magnitude’s average and its standard deviation, the area under the curve and the differences in magnitude between neighboring segments.

For each of the 36 GSAD data pairs, 16 features are extracted. These correspond to steady-state and transient characteristics, which evaluate the ascending and descending response of the sensor [47,48].

For each of the 40 GFE data pairs, three types of features are identified from the landmarks: distance, angle, and depth. From each frame, 21 characteristics are obtained: six distances and 12 depths, corresponding to 12 reference points, and three angles corresponding to nine landmarks. These 21 characteristics are then concatenated. Depending on the type of emotion, a different number of consecutive frames can be concatenated, which varies from two for Affirmative cases, Emphasis or Yes/No questions. Three are used for Conditional, four for Topic, five for Doubt, and six for Relative and Wh-questions.

### 4.3. Selecting the Best Set-Up of Fusion Architectures

The procedure to identify the best fusion architecture (see Section 3.1) for each dataset described above is as follows: First, each fusion architecture is tested repeatedly (24 times, following Demvsar [49]) with each dataset, recording the accuracy. Second, a Friedman test [36] is used to look for significant differences (95% confidence level) between pairs of configurations, considering the accuracy. Third, the Holm post hoc test [37] is used to test the difference (95% confidence level) between the aggregation architecture (taken as a reference) and the other architectures, based on the results of the Friedman test. Finally, with the results of the previous step, the fusion method with the best performance was selected for each of the 116 datasets studied here. We report the “best” fusion method for a particular row of the dataset as follows: in case no other method gets a better accuracy than aggregation, then aggregation is reported as the best, even if there is one or even several methods with better accuracy, but without a statistically significant difference. If one or several methods get better accuracy results than aggregation with a statistically significant difference, then, the one with the highest accuracy is reported as the best.

### 4.4. Predicting the Best Configuration of Fusion Architectures

The procedure to find the most suitable fusion architecture for a given dataset that belongs to any of the studied domains (SHA, GSAD, and GFE) consists of three stages. (a) A meta-dataset of SS is constructed for each of the three domains considered here. (b) The three meta-datasets are joint and their classes balanced. Furthermore, finally, (c) train a RFC-based classifier to identify the best fusion architecture. These stages are outlined below.

(a) To build the SS meta-dataset, first extract the SS (see Section 3.2) for each of the features (see Section 4.2) from the datasets for the three domains discussed here (see Section 4.1). The next step is to build a meta-dataset per domain, whose rows are the concatenated SSs of the features of the datasets of the corresponding domain. Then, because the number of columns (for the considered features) are different in each of the three meta-datasets, we reduced them using the SFFS algorithm (see Section 2.2) and the T transform (see Equation (Equation 8)) to be the same size. The SFFS algorithm was configured to reduce the features of the meta-datasets in a range between 3 and 20, using RFC as the classifier and the accuracy metric as the evaluation criterion, to find the best combination of features (sets between 3 and 20). The values of this metric were obtained through a three-fold cross-validation strategy with previously shuffled samples.

Therefore, from SFFS, we obtained a subset of nine features for the SHA meta-dataset ($f_{SHA}$), a subset of three characteristics for the Gas meta-dataset ($f_{Gas}$), and a subset of three features for the GFE meta-datasets ($f_{GFE}$). We chose these subsets because they achieved an accuracy of at least 90% when RFC was used with them: 91% accuracy with $f_{SHA}$, 92.5% accuracy with $f_{Gas}$, and 96.9% accuracy with $f_{GFE}$.

With the objective that the number of features is the same in the three meta-datasets, we applied the transformation *T* (defined in Equation (Equation 8)) to the features obtained above. We can see the result of this application in Equation (Equation 9), where $T:R^{m}\to R^{15}$ and m=9 or m=3.
(9)$$T\left(F\right)=\left\{\begin{array}{cccc} \left(f_{SHA},0,0,0,0,0,0\right) & \mathrm{if}F=f_{SHA}, & \mathrm{where} & f_{SHA}\in R^{9}\\ \left(0,0,0,0,0,0,0,0,0,f_{Gas},0,0,0\right) & \mathrm{if}F=f_{Gas}, & \mathrm{where} & f_{Gas}\in R^{3}\\ \left(0,0,0,0,0,0,0,0,0,0,0,0,f_{GFE}\right) & \mathrm{if}F=f_{GFE}, & \mathrm{where} & f_{GFE}\in R^{3} \end{array}\right.$$

The labels for each row of the three meta-datasets correspond to the best fusion architecture (identified in Section 4.3): MVSWSF, MVSWoSF, VotWSF, ABWRFC, or Agg. In the case of Agg, it is selected as the best when the difference with the others is not statistically significant.

(b) The three meta-datasets are joined, by row, to create a larger set with SS and labels that correspond to the best fusion architecture. We have made this large set of SS available at https://data.mendeley.com/datasets/vpjt5v26tc/1 (accessed on 3 October 2021). This dataset is larger and needs to be reviewed to ensure that there is a balance between classes. In this case, we use the up-sampling strategy, in which the samples from minority classes are increased by resampling with replacement to equalize the majority class [45,46]. The final characteristics of the balanced data corresponding to the SS are shown in Table 5.

(c) Both the RFC-based classifier and the large set of SS, created in the previous stage (b) (from now on, the meta-dataset of SS), are used to find the most suitable fusion architecture for a given dataset that fits any of the domains studied here. In this case, a tree folds cross-validation is used. The performance results of the classification and the comparison with the approach proposed in Brena et al. [12] are presented in Table 6. We use the precision, recall, f1-score, and accuracy metrics to measure the performance of our proposal and compare it to other works because these metrics have been commonly used for the same purposes [11,12]. Furthermore, we compare our approach only with Brena et al. [12]’s proposal since, as far as we know, they are the only ones tackling the same problem as us: finding the best fusion method for a given set of sensors for a particular set of domains. Although Aguileta et al. [11] addressed the same topic, they only work on a specific domain.

## 5. Results and Discussion

After balancing the number of classes in each of the sets, Table 5 displays the characteristics of the SS dataset. The dimensions are 235 rows that correspond to the 47 instances (after upsampling) belonging to each of the 5 fusion architectures. The 16 columns correspond to the 15 reduced features (see Equation (Equation 9)) and the label. We emphasize the balance of the classes, which will favor a fair measurement in metrics such as accuracy.

Table 6 shows the details of identifying the most suitable setting of the considered fusion architectures.

In Table 6, we can see that the three metrics (precision, recall, and f1-score) reached values between 90% and 100% with MVSWSF, VotWSF, and MVSWoSF, whereas these metrics reached a value of 85% for Agg. This table also shows the support which is the number of examples per class. These observations give evidence that the RFC classifier can predict MVSWoSF, VotWSF, and MVSWSF well, and reasonably well Agg when trained with the SS dataset created with SFFS.

When contrasting the approach proposed by Brena et al. [12] with our work, from the perspective of the average value presented by the metrics in Table 6, we can see that the performance achieved using the RFC classifier in combination with the SFFS is 93% (on average for all metrics), which is higher than the 91% (on average for all metrics) performance achieved by the combined RFC classifier and the PCA. This result suggests that the SS set reduced by the SFFS algorithm contains more informative, discriminatory, and independent characteristics (which the RFC could take advantage of) than the SS set reduced by the PCA method.

Even though the SS dataset built with SFFS helped RFC produce better performance, it took more steps to create it (calculate a subset of more relevant features and change the size of this subset, see step 1 of Section 4.4) than the steps necessary to create the SS dataset using PCA (extract the PCA components [14] and then combine then by row [12]). Therefore, a user who uses our method (with the SS dataset built with PCA) would obtain a quite good accuracy (91%) in the task of recognizing the best fusion architecture for a given dataset, in a more direct way.

On the other hand, because PCA transforms the data assuming that they maintain a linear relationship, and it is not always the case, the most general way to create the SS dataset is to use the SFFS algorithm, since this algorithm takes a subset of features (without transforming them, see step 1 of Section 4.4) that helps a classifier achieve the best performance (see Table 6).

From the results obtained in Table 6, we see that it is possible to estimate the most appropriate fusion architecture, based on the average value of the performance metrics for each dataset. Likewise, the transformation of the data described in Equations (Equation 8) and (Equation 9) allows organizing the individual sets in such a way that they all contain the same number of characteristics. These results suggest that the proposed methodology may be useful in data-driven supervised learning applications whose operating environment (sensors) is constantly changing. For example, applications for the domains here presented (SHA, GSAD, and GFE). With our approach, these applications could react to changes in sensor data and constantly change the best way to fuse that data, based on this changing data, to maintain the best possible performance on its task (prediction or classification).

## 6. Conclusions

In this article, we describe, test and assess the use of the SFFS algorithm and a structural transformation (the “*T* transformation”) in the construction of a statistical signature dataset for several domains (demonstrated here with three different ones), resulting in an accuracy improvement in the task of predicting the best performing fusion architecture for a given dataset. An important consideration when using our proposal arise while building the SS meta-dataset, with each domain added, the number of columns (features) increases, causing the *T* transformation to be redefined and applied in each domain each time. This does not happen when using PCA, since in this method the number of columns does not change, even in cases where the number of domains does change. Finally, our proposal can benefit applications based on data-driven supervised learning algorithms that require maintaining the best possible performance in the face of changes in their environments. Our proposal can help these applications to identify the best fusion method according to the changes in the data and thus increase the possibility of maintaining or even improving, the performance in the tasks of these applications to classify or to predict.

## Figures and Tables

**Figure 1 sensors-21-07007-f001:**
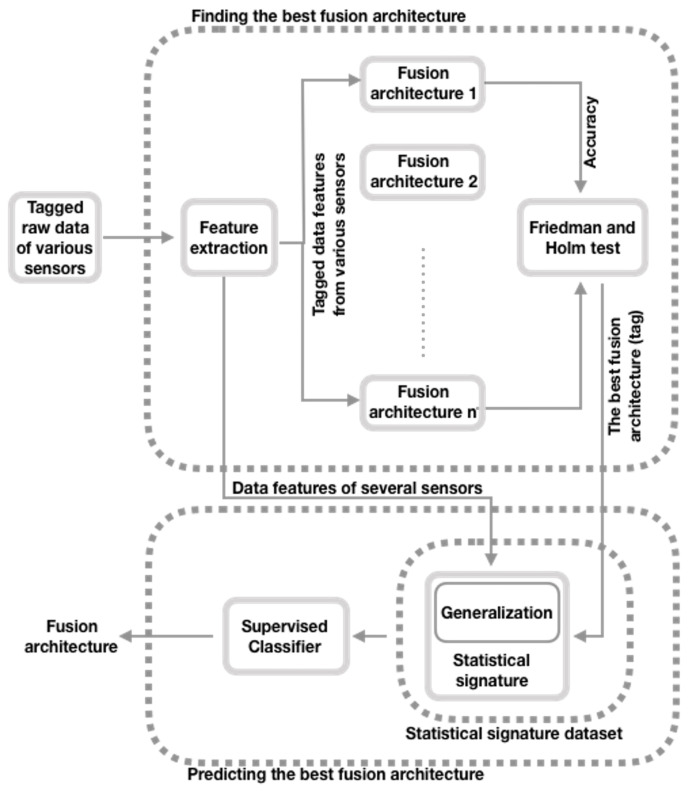
General schematic of the enhanced approach that identifies the optimal fusion architecture (adapted from Brena et al. [12]).

**Figure 2 sensors-21-07007-f002:**
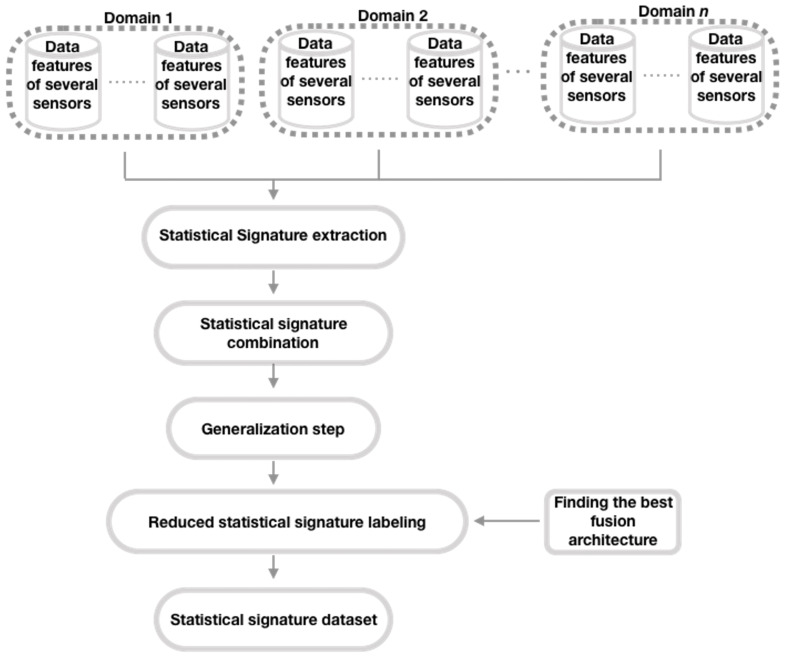
Steps to generate the SS database (adapted from Brena et al. [12]).

**Figure 3 sensors-21-07007-f003:**
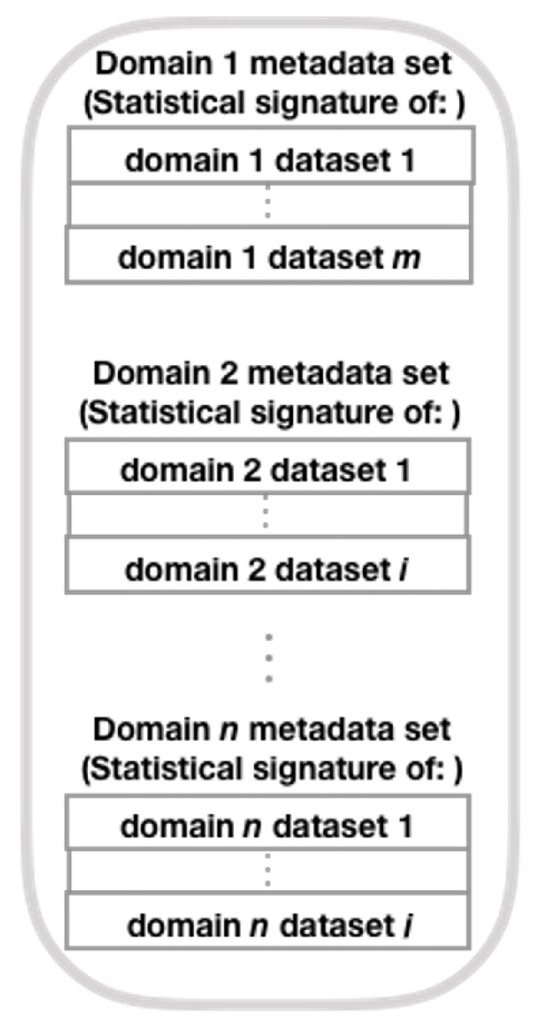
Statistical signature combination (adapted from Brena et al. [12]).

**Figure 4 sensors-21-07007-f004:**
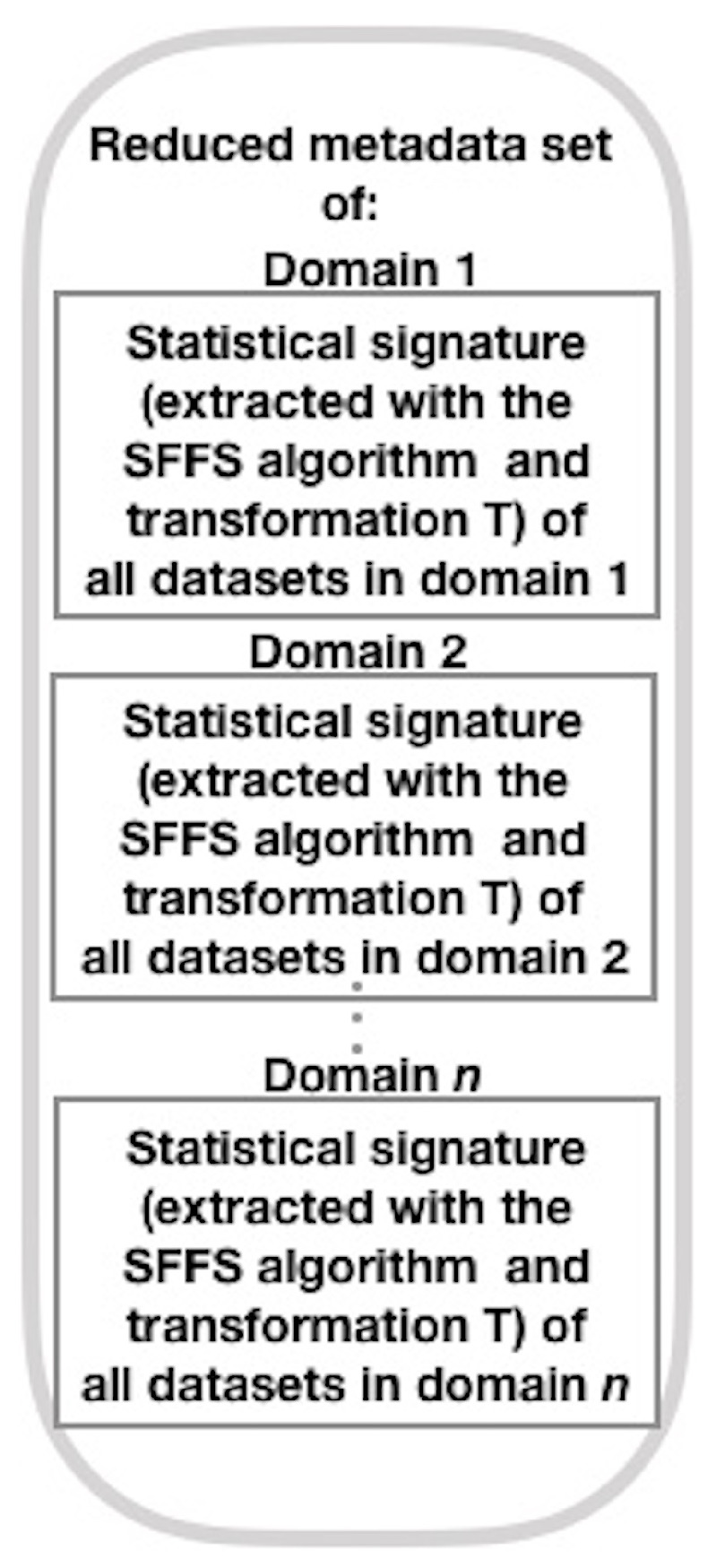
Generalization step (adapted from Brena et al. [12]).

**Figure 5 sensors-21-07007-f005:**
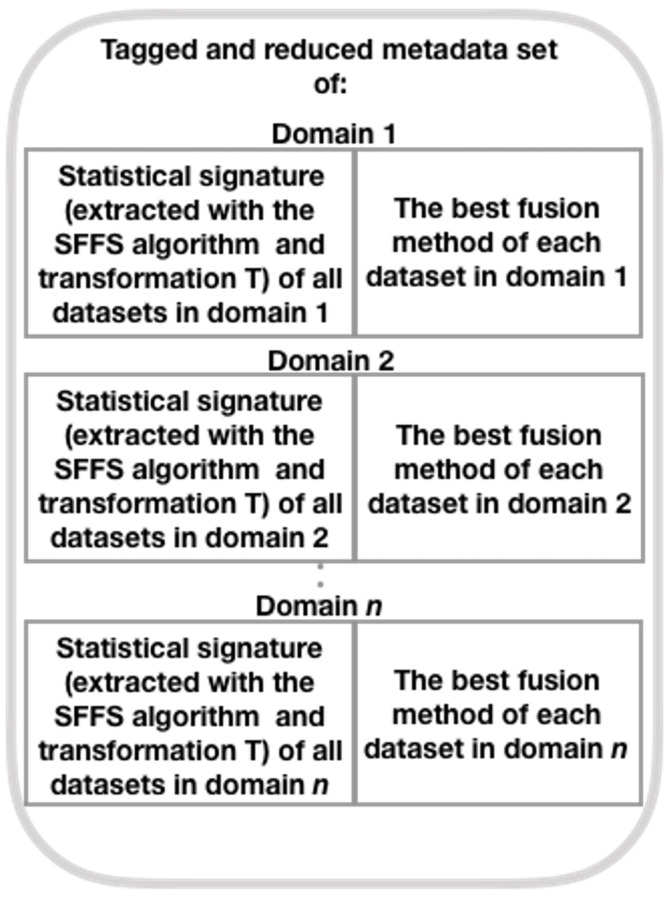
Reduced statistical signature labeling (adapted from Brena et al. [12]).

**Figure 6 sensors-21-07007-f006:**
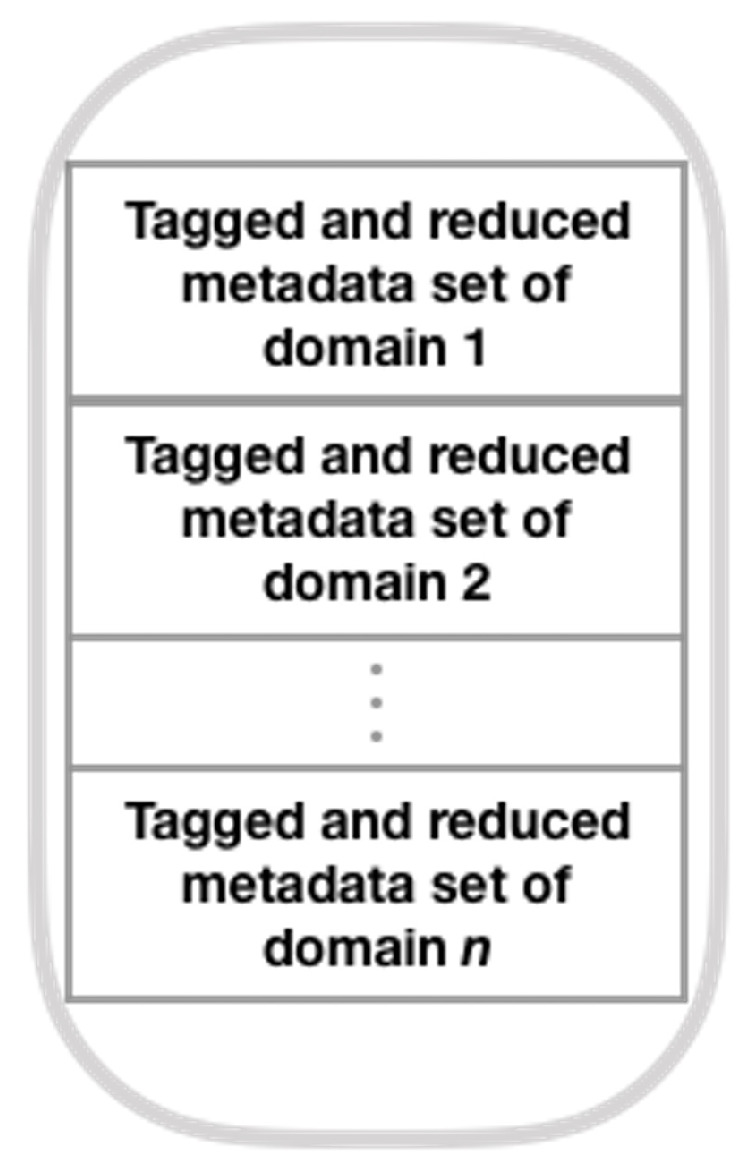
Statistical signature dataset (adapted from Brena et al. [12]).

**Table 1 sensors-21-07007-t001:** Sensor pairs used to derive the Opportunity datasets. Ba = Back, Rl = Right lower arm, Ru = Right upper arm, Lu = Left upper arm, and Ll=Left lower arm.

Sensor Pairs	Description
1	Acc and Gyr of the Rl
2	Ba Acc and Ll Gyr
3	Ba Acc and Lu Gyr
4	Ba Accr and Rl Gyr
5	Ba Acc and Ru Gyr
6	Ll Acc and Ba Gyr
7	Acc and Gyr of the Ll
8	Acce and Gyr of the Ru
9	Ru Acc and Ll Gyr
10	Ru Acc and Lu Gyr

**Table 2 sensors-21-07007-t002:** Sensor used to extract the PAMAP2 datasets. Ha = Dominant arm, Ch = Chest, and An = Dominant side’s ankle.

Sensor Pairs	Description
1	Acc and Gyr of the Ha
2	Acc and Gyr of the An
3	An Acc and Ha Gyr
4	Acc and Gyr of the Ch
5	Ch Acc and Ha Gyr
6	Ha Acc and An Gyr
7	Ha Acc and Ch Gyr

**Table 3 sensors-21-07007-t003:** Sensor used to derive the Mhealth datasets. Ch = Chest, Ra = Right lower arm (Ra), and La = Left ankle.

Sensor Pairs	Description
1	Acc and Gyr of the Ra
2	Acc and Gyr of the La
3	La Acc and Ra Gyr
4	Ra Acc and La Gyr

**Table 4 sensors-21-07007-t004:** Sensor used to extract the DSA data sets. To = Torso, Ra = Right arm, La = Left arm, Rl = Right leg, and Ll = Left leg.

Sensor Pairs	Description
1	La Acc and Ll Gyr
2	La Accr and Rl Gyr
3	Ll Acc and La Gyr
4	Ll Acc and Ra Gyr
5	Ll Acc and Rl Gyr
6	Ra Acc and Rl Gyr
7	Rl Acc and La Gyr
8	Rl Acce and Ll Gyr
9	Rl Acc and Ra Gyr
10	Rl Acc and To Gyr
11	Acc and Gyr of the Ra
12	Acc and Gyr of the La
13	Acc and Gyr of the Ll
14	Acc and Gyr of the Rl
15	Acc and Gyr of the To
16	To Acc and Ll Gyre
17	To Acc and Ra Gyr

**Table 5 sensors-21-07007-t005:** Final SS dataset.

Dataset	Number of (Rows, Columns)	Division of Classes				
		Agg	MVSWFS	VotWSF	MVSWoFS	ABWRFC
SS	(235, 16)	47	47	47	47	47

**Table 6 sensors-21-07007-t006:** Fusion strategies identification (RFC-based performance results).

Label	Precision		Recall		f1-Score		Support
	Ours	Brena [12]	Ours	Brena [12]	Ours	Brena [12]	
ABWRFC	0.98	1.00	1.00	1.00	0.99	1.00	47
Agg	0.85	0.87	0.85	0.87	0.85	0.87	47
MVSWSF	0.91	0.88	0.91	0.81	0.91	0.84	47
MVSWoSF	0.90	0.90	0.96	0.91	0.93	0.91	47
VotWSF	1.00	0.92	0.91	0.98	0.96	0.95	47
avg/total	0.93	0.91	0.93	0.91	0.93	0.91	235
accuracy					0.93	0.91	235

## Data Availability

Data generated for this study is publicly available at: https://data.mendeley.com/datasets/vpjt5v26tc/1 (accessed on 1 October 2021).

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
