# Peer review of "Improved Accuracy in Predicting the Best Sensor Fusion Architecture for Multiple Domains"

_sensors, 2021, doi:10.3390/s21217007_

Round 1

Reviewer 1 Report

author try to build algorithm to implement the optimal fusion architecture. the topic is worthy for the engineer in this special field, but some important issues need to be responded before published .
1) Please describe the architecture flow and the responding results clearly by adding the curve and comparing analysis.
2) "best" represents the top one. please add some other examples to prove 'best'.   
3) is the proposed method used commonly to fusion any multi sensors or physical information?

Author Response

General Comment: Authors try to build an algorithm to implement the optimal fusion architecture. The topic is worthy for the engineer in this special field, but some important issues need to be responded before published.

Answer: We thank Reviewer #1 for the interest in the manuscript. Specific comments are discussed below.

Comment 1: Please describe the architecture flow and the responding results clearly by adding the curve and comparing analysis.

Answer 1: Yes. We describe the general architecture in Section 3 and in Figure 1. We have improved the following paragraph in order to make it more clear. (lines 174-182)

Figure 1 presents the approach  proposed by Brena et al. [12] that predicts the most suitable fusion architecture for a given set of data (e.g., HAR [32], gas detection [13], or GFE recognition [6]). In this section we briefly explain this approach, first, in subsection 3.1  the process for choosing the most effective fusion architecture is described. Then, in subsection 3.2, the details of the statistical signature dataset stage are explained, as this is the core of our proposal, and we are following a different approach than the one described by the original authors. And finally, in section 3.3, the prediction of the optimal fusion architecture step is briefly described. For more details, the reader can consult the work of Brena et al. [12].”

Also, in order to help in the description of the architecture, we have included new figures (2 to 6). 

Regarding the results analysis, a comparative analysis is presented in Table 6 and in Section "5. Results and Discussion". Please, see lines (441 and 442-447).

Comment 2:  "best" represents the top one. Please add some other examples to prove 'best'.

Answer 2:We clarified exactly what we meant by “best” at the end of section 4.3, lines 385-386 in the original paper, like this:
We report the “best” fusion method for a particular row of the dataset as follows: in case no other method gets a better accuracy than aggregation, then aggregation is reported as the best, even if there is one or even several methods with better accuracy, but without a statistically significant difference. If one or several methods get better accuracy results than aggregation with a statistically significant difference, then, the one with the highest accuracy is reported as the best.

You can find examples of this selection of the “best” fusion method in our previous work (reference 12 in our current paper); we avoided repeating them in the current paper.

Comment 3: Is the proposed method used commonly to fusion any multi sensors or physical information?

Answer 3: It is a new method that is currently in the process of being presented to the scientific community. We believe that it can help in tasks of data fusion and we hope it will be used in the future.

Comment 4: From the questionnaire (“Moderate English changes required” was checked).

Answer 4: We did the following language corrections (we first indicate the line number in the original paper, then the phrase to correct, and then the corrected version):

Line 4: “, and then” changed to “, so”

Line 4: “difficult” changed to “challenging”

Line 28: “certain” changed to “specific”

Line 28: “particularly” changed to “mainly”

Line 36: “approach” changed to “one”

Line 71: “by means of implementing” changed to “through”

LIne 80: “a background on the state of the art” changed to “a review of the state-of-the-art”

Line 84: “set up” changed to “setup”

Line 111: “seen in Dynamic multi-sensor data fusion approach” changed to “seen in the dynamic multi-sensor data fusion approach”

Line 112: “operato” changed to “operator”

Line 113: “two-levels” changed to “two levels”

Line 120 “without further evaluation, consequently...” changed to “without further evaluation. Consequently...”

Line 139: “algorithm, from [15]” changed to “algorithm, taken from Pudil et al. [15]”

Line 174: “In this Section” changed to “In this section,”

Line 177: “proposal and we are” changed to “proposal, and we are”

Line 182: “out from eight predefined ones” changed to “out of eight predefined ones”

Line 189: “every available features” changed to “every available feature”

Line 192: “true labels” changed to “actual labels”

Line 201: “(shuffle the features)” changed to “(to shuffle the features)”

Line 234: “in order to it identifies” changed to “so that it identifies”

Line 237: “with respect to accuracy” changed to “concerning accuracy”

Line 239: “In the case the accuracy of some fusion...” changed to “Some fusion...”

Line 241: “...the best of the examined fusion architectures is taken as such...” changed to “The best of the examined fusion architectures is taken as such, as explained above.”

Line 252: “...which we conclude is a simplified set of meta-data...” changed to “..., which is a simplified set of meta-data...”

Line 273: “optimal performance” changed to “maximum performance”

Line 274: “determined by the criterion function” changed to “determined by the following criterion”

Line 285: “labelled” changed to “labeled”

Line 286: “In order to crate the final dataset  of SS, the labelled…” changed to “To create the final dataset  of SS, the labeled”

Line 292: “cross validation” changed to “cross-validation”

Line 295: “Set Up” changed to Set-Up”

Line 319: “as” changed to “like”

Line 328: “The original dataset consists on activity...” changed to “The original dataset consists of activity...”

Line 337: “The original data consist on sports activities...” changed to “The original data consist of sports activities...”

Line 347: “From the data of the month 36...” changed to “From the data of month 36”

Line 354: “These cover each eyes...” changed to “These cover each eye...”

Line 398: “(for the now considered features)” changed to “(for the considered features)”

Line 423: “increased by resample” changed to “increased by resampling”

Line 475: “...allows to organize the individual sets...” changed to “...allows organizing the individual sets...”

Line 487: “An important consideration using our proposal arise...” changed to “An important consideration when using our proposal arises...”

Reviewer 2 Report

Multi-sensor fusion aims to improve the overall reliability during the decision-making process, or to allow one sensor to compensate for the shortcomings of other sensors. This field has received much attention and research. Various researchers have proposed different fusion methods, or called as “architecture".  However, it is difficult to specify which fusion method or architecture is more suitable for a particular setup of sensor platform or a specific application environment except through trial and error. The authors proposed a method that, from predefined options, can predict the best fusion architecture for a given data set. The method involves constructing a metadata set in which statistical features are extracted from the original data set. One challenge is that each data set has a different number of variables or columns. Previous work proposed in the literature used the first k components of principal component analysis to make the metadata set column coherent and machine learning classifiers are trained to predict the best fusion architecture. The authors of the paper take a different approach to building metadata sets. The Sequential Forward Floating Selection algorithm and a T Transform are used to reduce the features and match them to a predefined number respectively. The findings indicate that the proposed method can improve the accuracy of predicting the best sensor fusion architecture of multiple domains.

The paper is generally well written. The illustration of Figure 2. can be improved by separating the diagrams into two or three diagrams. The fonts used in Figure 2. seem a bit small. 

Author Response

General Comment: Multi-sensor fusion aims to improve the overall reliability during the decision-making process, or to allow one sensor to compensate for the shortcomings of other sensors. This field has received much attention and research. Various researchers have proposed different fusion methods, or called as “architecture".  However, it is difficult to specify which fusion method or architecture is more suitable for a particular setup of sensor platform or a specific application environment except through trial and error. The authors proposed a method that, from predefined options, can predict the best fusion architecture for a given data set. The method involves constructing a metadata set in which statistical features are extracted from the original data set. One challenge is that each data set has a different number of variables or columns. Previous work proposed in the literature used the first k components of principal component analysis to make the metadata set column coherent and machine learning classifiers are trained to predict the best fusion architecture. The authors of the paper take a different approach to building metadata sets. The Sequential Forward Floating Selection algorithm and a T Transform are used to reduce the features and match them to a predefined number respectively. The findings indicate that the proposed method can improve the accuracy of predicting the best sensor fusion architecture of multiple domains.

The paper is generally well written. 

Answer: We thank the reviewer for the attentive reading of the article. Specific comments are discussed below.

Comment 1: The illustration of Figure 2. can be improved by separating the diagrams into two or three diagrams. The fonts used in Figure 2. seem a bit small.

Answer 1: Yes, we agree. We have improved the diagram from the original Figure 2 by dividing it into five diagrams (see Figures 2, 3, 4, 5, and 6). Also, the font size has been increased in those Figures. Please see Figures 2 to 6.

Comment 2: “English language and style are fine/minor spell check required” was checked.

Answer 2: Yes, we agree. We did the following language corrections (we first indicate the line number in the original paper, then the phrase to correct, and then the corrected version):

Line 4: “, and then” changed to “, so”

Line 4: “difficult” changed to “challenging”

Line 28: “certain” changed to “specific”

Line 28: “particularly” changed to “mainly”

Line 36: “approach” changed to “one”

Line 71: “by means of implementing” changed to “through”

LIne 80: “a background on the state of the art” changed to “a review of the state-of-the-art”

Line 84: “set up” changed to “setup”

Line 111: “seen in Dynamic multi-sensor data fusion approach” changed to “seen in the dynamic multi-sensor data fusion approach”

Line 112: “operato” changed to “operator”

Line 113: “two-levels” changed to “two levels”

Line 120 “without further evaluation, consequently...” changed to “without further evaluation. Consequently...”

Line 139: “algorithm, from [15]” changed to “algorithm, taken from Pudil et al. [15]”

Line 174: “In this Section” changed to “In this section,”

Line 177: “proposal and we are” changed to “proposal, and we are”

Line 182: “out from eight predefined ones” changed to “out of eight predefined ones”

Line 189: “every available features” changed to “every available feature”

Line 192: “true labels” changed to “actual labels”

Line 201: “(shuffle the features)” changed to “(to shuffle the features)”

Line 234: “in order to it identifies” changed to “so that it identifies”

Line 237: “with respect to accuracy” changed to “concerning accuracy”

Line 239: “In the case the accuracy of some fusion...” changed to “Some fusion...”

Line 241: “...the best of the examined fusion architectures is taken as such...” changed to “The best of the examined fusion architectures is taken as such, as explained above.”

Line 252: “...which we conclude is a simplified set of meta-data...” changed to “..., which is a simplified set of meta-data...”

Line 273: “optimal performance” changed to “maximum performance”

Line 274: “determined by the criterion function” changed to “determined by the following criterion”

Line 285: “labelled” changed to “labeled”

Line 286: “In order to crate the final dataset  of SS, the labelled…” changed to “To create the final dataset  of SS, the labeled”

Line 292: “cross validation” changed to “cross-validation”

Line 295: “Set Up” changed to Set-Up”

Line 319: “as” changed to “like”

Line 328: “The original dataset consists on activity...” changed to “The original dataset consists of activity...”

Line 337: “The original data consist on sports activities...” changed to “The original data consist of sports activities...”

Line 347: “From the data of the month 36...” changed to “From the data of month 36”

Line 354: “These cover each eyes...” changed to “These cover each eye...”

Line 398: “(for the now considered features)” changed to “(for the considered features)”

Line 423: “increased by resample” changed to “increased by resampling”

Line 475: “...allows to organize the individual sets...” changed to “...allows organizing the individual sets...”

Line 487: “An important consideration using our proposal arise...” changed to “An important consideration when using our proposal arises...”